Selective HDAC6 inhibitor TubA offers neuroprotection after intracerebral hemorrhage via inhibiting neuronal apoptosis

Peng Cuiying
Gong Xiyu
Hu Zhiping
Chen Chunli 168202083@csu.edu.cn
Jiang Zheng zhengjiang187@csu.edu.cn
Department of Neurology, Second Xiangya Hospital, Central South University , Changsha, Hunan , China
Xu Yuzhen
Electronic publication date: 2023 Apr 28
Publication date: 2023
Volume: 11
Electronic Location ID: e15293
Received 2023 Jan 16; Accepted 2023 Apr 4
Copyright: © 2023 Peng et al.
Copyright year: 2023
Copyright holder: Peng et al.
License: This is an open access article distributed under the terms of the Creative Commons Attribution License, which permits unrestricted use, distribution, reproduction and adaptation in any medium and for any purpose provided that it is properly attributed. For attribution, the original author(s), title, publication source (PeerJ) and either DOI or URL of the article must be cited.
License URL: https://creativecommons.org/licenses/by/4.0/

Keywords: HDAC6, Acetylation of α-tubulin, Tubastatin A, Intracerebral hemorrhage, Apoptosis

Funding: National Natural Science Foundation of China 81974212 Natural Science Foundation of Hunan Province 2021JJ40821 This work was supported by the National Natural Science Foundation of China (Grant no. 81974212) and the Natural Science Foundation of Hunan province (Grant no. 2021JJ40821). The funders had no role in study design, data collection and analysis, decision to publish, or preparation of the manuscript.

==============================
A large body of evidence has demonstrated that neuronal apoptosis is involved in the pathological process of secondary brain injury following intracerebral hemorrhage (ICH). Additionally, our previous studies determined that the inhibition of HDAC6 activity by tubacin or specific shRNA can attenuate neuronal apoptosis in an oxygen-glucose deprivation reperfusion model. However, whether the pharmacological inhibition of HDAC6-attenuated neuronal apoptosis in ICH remains unclear. In this study, we used hemin-induced SH-SY5Y cells to simulate a hemorrhage state in vitro and adopted a collagenase-induced ICH rat model in vivo to assess the effect of the HDAC6 inhibition. We found a significant increase in HDAC6 during the early stages of ICH. As expected, the acetylated α-tubulin significantly decreased in correlation with the expression of HDAC6. Medium and high doses (25, 40 mg/kg) of TubA, a selective inhibitor of HDAC6, both reduced neurological impairments, histological impairments, and ipsilateral brain edema in vivo. TubA or HDAC6 siRNA both alleviated neuronal apoptosis in vivo and in vitro. Finally, HDAC6 inhibition increased the level of acetylated α-tubulin and Bcl-2 and lowered the expression of Bax and cleaved caspase-3 post-ICH. In general, these results suggested that the pharmacological inhibition of HDAC6 may act as a novel and promising therapeutic target for ICH therapy by up-regulating acetylated α-tubulin and reducing neuronal apoptosis.

Introduction

Intracerebral hemorrhage (ICH), a common clinically acute cerebrovascular disease that is characterized by primary non-traumatic hemorrhage in the brain parenchyma, is a catastrophic stroke subtype with high morbidity and mortality (van Asch et al., 2010; Xi, Keep & Hoff, 2006). Despite recent progress in the treatment of ICH, roughly 20% of ICH cases recover limb function half a year post-ICH (Hemphill et al., 2015). The cascade brain injury triggered by ICH is usually occurs in two consecutive pathological courses. In the first stage of brain injury, the hematoma of the ruptured vessel constructs a cascading mass effect that mechanically stretches and destroys adjacent cerebral tissue. The second stage of brain injury, which is mainly initiated by the degradation products of blood cells and blood components, includes microglia activation, oxidative stress, inflammation, blood-brain barrier (BBB) disruption, and cerebral edema. This will lead to neuronal apoptosis and severe neurological deficits (Qureshi, Mendelow & Hanley, 2009; Wilkinson et al., 2018). A large body of evidence has indicated that inhibiting neuronal apoptosis could attenuate ICH and ameliorate the neural function deficit score after ICH (Pan et al., 2018; Xu et al., 2018). Therefore, more efforts are required to probe the mechanisms of apoptosis and explore novel therapeutic strategies targeting neuronal apoptosis in order to improve the prognosis of ICH patients.

In recent years, histone deacetylase inhibitors (HDACs) have been widely used as a new therapy for stroke and neurodegenerative diseases via various mechanisms (Wang, Fessler & Chuang, 2011). Pan and different isoform-specific HDAC pharmacologic inhibitors exhibited favorable influences in different cerebral diseases including hemorrhagic stroke (Sukumari-Ramesh, Alleyne & Dhandapani, 2016). Among 18 identified HDACs in mammals, HDAC6 is a characteristic member of the class IIb subgroup that is mainly located in the cytoplasm and has two catalytic domains (Hubbert et al., 2002). Tubastatin A (TubA), the most effective inhibitor against HDAC6, has an IC50 of 15 nM and a selectivity over 1,000-fold for all HDAC isoforms apart from HDAC8 (Butler et al., 2010). Notably, recent studies have found that HDAC6 inhibition resulted in brain protection in several types of central nervous system disorders including Alzheimer’s disease, Huntington’s disease, and ischemia stroke (Wang et al., 2016; Zhang et al., 2019, 2014). Additionally, our previous study demonstrated that the inhibition of HDAC6 activity by tubacin or specific shRNA inhibited caspase-3 activation and attenuated neuronal apoptosis in an oxygen-glucose deprivation reperfusion model (Zhang et al., 2019). However, whether pharmacological inhibition or gene interference of HDAC6 attenuated neuronal apoptosis in cellular or animal ICH models remains unclear. Therefore, this work was designed to investigate the role of HDAC6 during the course of ICH induction and the protective effects of TubA on apoptosis in an ICH model.

Materials and Methods

In vivo study

Animals and experimental groups

One hundred and ninety-six adult male Sprague-Dawley rats (250–350 g) purchased from Hunan Slack Jingda Animal Laboratory (Changsha, China) were used for this experimental research. Sprague–Dawley rats were raised three per cage in the central laboratory of Hunan Provincial People’s Hospital at a fixed temperature (25 °C) and relative humidity (60%). The rats were fed with free water and food, and dwelt in a 12-h light/dark cycle condition. Animals were stochastically allocated to each experiment group. All experimental programs were authorized by the Institutional Animal Care and Use Committee of Central South University (IACUC approval No: 2020079). The SD rats were stochastically divided into four experimental groups: the sham-treated group (no ICH, control group), vehicle group (ICH + DMSO), low dosage treatment group (ICH + TubA 25 mg/kg), and high dosage group (ICH + TubA 40 mg/kg). All rats were euthanized humanely using an intraperitoneal injection of pentobarbital sodium.

Rat model of ICH

The ICH procedure was performed by injecting collagenase type IV (Sigma-Aldrich, St. Louis, MO, USA) through stereotaxic intrastriatal implantation, as previously described with some small alterations (Rosenberg et al., 1990; Zeng et al., 2017). In simple terms, animals were deeply anesthetized with pentobarbital sodium (40 mg/kg) via intraperitoneal injection. Under deep anesthesia, the SD rats were placed carefully in the stereotactic apparatus and an incision was made along the midline to expose the bregma. We stereotactically inserted a microsyringe (10 µL) into the right striatum through the cranial drill hole with coordinates of 0.2 mm (anterior–posterior), 3 mm (right, medial–lateral), and 6 mm ventral (from pia, dorsal–ventral (DV)) from the bregma. Collagenase IV (0.2U) dissolved in 2 μl saline was injected slowly for 5 min, placed for another 5 min, and then gradually removed. The rats’ body temperatures were maintained at 37 ± 0.5 °C using a homoeothermic blanket control unit. The sham operation was performed using all the above-mentioned procedures but only an equal volume of saline was injected.

Experimental design

The allocation and use of rats is illustrated in Table S1.

Experiment I: time-course of HDAC6 (n = 30)

The variations of HDAC6 protein levels in perihematomal tissues over time were observed. Thirty rats were stochastically assigned into six experimental groups: sham, ICH 4 h, ICH 6 h, ICH 12 h, ICH 1 d, and ICH 3 d. At each time point post-ICH, three rats were used for Western blot (WB) assay.

Experiment II: effect of Tubastatin A (n = 92)

We investigated the effect of TubA on neuronal injuries induced by ICH. Rats (n = 92) were stochastically allocated into four groups: sham, ICH+ vehicle, ICH+TubA 25 mg/kg, and ICH+TubA 40 mg/kg. We determined the neurological scores and calculated the brain water content at 1 d and 3 d after operation in each group (n = 10/group). Western blot assay was conducted in each group (n = 3) to quantitatively analyze the level of HDAC6 at 3 d after ICH.

Experiment III: mechanisms of the TubA effect (n = 48)

We attempted to further probe the mechanisms of the neuroprotective effect executed by TubA. First, for the sham, ICH+ vehicle, ICH+TubA 25 mg/kg, and ICH+TubA 40 mg/kg groups, we applied Western blot assay to explore the level of acetylation α-tubulin in each group (n = 3). Second, the protein levels of Bcl-2, Bax, and cleaved caspase-3 were also detected using Western blot assay (n = 3/group) in each group. Brain sections from the sham (n = 6), ICH+ vehicle (n = 6), ICH+TubA 25 mg/kg (n = 6), and ICH+TubA 40 mg/kg (n = 6) groups were prepared for HE staining and TUNEL staining.

Drug administration

Tubastatin A (TubA, #A4101; ApexBio Technology, Houston, TX, USA) was first dissolved in 4% dimethylsulfoxide (DMSO) and then diluted with 40% Polyethylene glycol 300 (PEG300) in double-distilled water (ddH2O). TubA was injected intraperitoneally 30 min before ICH induction with two dosages: 25 and 40 mg/kg, according to previous studies (Wang et al., 2016). DMSO solution in ddH2O with PEG300 was injected in the sham group.

Behavioral testing

Behavioral testing was conducted at 1 and 3 d post-ICH in the light of the previous study (Garcia et al., 1995). The modified Garcia scoring system included six individual tests: voluntary movement, symmetry of limbs, forelimb extension, body proprioception, climbing ability, and tentacle touch. Each individual trial was scored from 0 to 3. The modified Garcia test scores ranged from 3 to 18. A minimum score of three points suggested severe behavior impairment and the highest score of 18 points represented proper neurological function. All trials, assessments, and computations were blindly monitored by two independent investigators.

Brain water content (BWC) assessment

The BWC of each group was measured at 1 and 3 d post-ICH, following the previous methods (Suzuki et al., 2010). The brains of rats were rapidly dissociated after deep anesthetization and sacrifice. The brains were then dissected into five sections: contralateral and ipsilateral basal ganglia, contralateral and ipsilateral cortex, and cerebellum. All parts were quickly weighed on a precision electronic autobalance for wet weight (WW) and dried at 100 °C for dry weight (DW). The calculation formula was: BWC = [(WW-DW)/WW] × 100%.

Hematoxylin and eosin staining

First, we perfused the rats with 4% paraformaldehyde, decapitated and collected the rat brains, fixed the rat brains in 4% paraformaldehyde for 1 day, paraffin-embedded the brains, and cut them into slices (4 μm thick). The sections were flushed with tap water and stained with hematoxylin for 5 min, differentiated using 0.1% hydrochloric acid and ethanol solution for 25 s, blued in phosphate buffered saline for 45 min, counterstained with Eosin for 5 min, dehydrated with 95% alcohol, and finally mounted with neutral balsam. Areas with intact or lysed red blood cells were identified as hematoma. The perihematomal images were obtained using an optical microscope (Nikon Eclipse E100).

Double staining of TUNEL and NeuN

To confirm neuronal apoptosis in perihematomal tissues, we performed double immunofluorescent staining of the neuron-specific nuclear protein (NeuN) with TdT-mediated dUTP-biotin nick end labeling (TUNEL). TUNEL staining was conducted following the manual of the Cell Apoptosis Detection Kit (G1501-50T; ServiceBio, Wuhan, China), followed by staining against the neuronal marker NeuN (1:200; Abcam, Cambridge, UK). Finally, the slides were blanketed with DAPI (Beyotime, Shanghai, China). The TUNEL-positive neurons were captured using a fluorescence microscope (Nikon Eclipse C1) and analyzed with Image J software.

In vivo Western blot

Perihematomal tissues from the ipsilateral hemisphere were homogenized and centrifuged, then separated using SDS-poly-acrylamide gel electrophoresis, and transmitted to nitrocellulose membranes. Next, we blocked the membrane with defatted milk (5%) for 1 h and incubated it overnight at 4 °C with the following primary antibodies: rabbit anti-HDAC6 antibody (1:2,000, #ab239362; Abcam, Cambridge, UK), mouse anti-ace-α-tubulin (1:2,000, #ab24610; Abcam, Cambridge, UK), rabbit anti-cleaved caspase-3 antibody (1:1,000, #9661; Cell Signaling Technology, Danvers, MA, USA), rabbit anti-Bax antibody (1:1,000, #ab32503; Abcam, Cambridge, UK), rabbit anti-Bcl-2 antibody (1:1,000, #ab196495; Abcam, Cambridge, UK), or mouse anti-β-actin antibody (1:5,000, #66009-1-Ig; Proteintech, Chicago, IL, USA). Corresponding secondary antibodies were chosen to incubate for 90 min. The data were analyzed using Image J software.

In vitro study

Cell culture and ICH model

The SH-SY5Y cells (American Type Culture Collection, Manassas, VA, USA) were cultured in Dulbecco’s modified Eagle’s medium (DMEM) (Corning, NY, USA), accompanied with FBS (10%, Gibco) and penicillin-streptomycin (1%; Gibco) at 37 °C with 5% CO2 in a moist environment. Treatment of 10 μM hemin (#C3984; ApexBio, TX, USA) for 24 h was used to induce an in vitro ICH model.

Cell viability: CCK-8 assay

Measurement of cell viability was determined using a CCK-8 detection kit (Apexbio, USA). In short, SH-SY5Y cells (1 × 104 cells/mL) were sowed in 96-well plates with culture medium (100 μl/well) for 24 h. Subsequently, a CCK-8 solution (10 μL) was mixed into each well. Absorbance was gauged at 450 nm with a Bio Tek microplate reader.

Drug administration

SH-SY5Y cells were pretreated with Tubastatin A (3 μM) dissolved in DMSO for 6 h as described previously (Yu et al., 2016), and then co-treated with 10 μM hemin (Goldstein et al., 2003) for an additional 24 h. The vehicle group added an equal volume of PBS.

HDAC6 knockdown in SH-SY5Y cells

HDAC6 siRNA or siRNA-NC was transfected into SH-SY5Y cells using Lipofectamine 3000 (L3000-015; Invitrogen, Waltham, MA, USA). After transfection for 48 h, SH-SY5Y cells were treated with hemin for 24 h and collected for further analysis.

Cell apoptosis: flow cytometry

The culture medium was discarded and SH-SY5Y cells were collected using EDTA-free trypsin, then rinsed twice with PBS. We centrifuged the suspension, obtained about 3.2 × 105 cells, and added 500 μL of binding buffer suspension to the collected SH-SY5Y cells. The suspension was blended with Annexin V-APC (5 μl) and we added PI (5 μl) to be reincubated for 10 min. Finally, we analyzed the mixture instantly with an FACS flow cytometer C6 (FCM; FACSCanto II; BD Biosciences) and the results were analyzed using FlowJo v10.7.1 (Tree Star, Ashland, OR, USA).

In vitro Western blot

We threw away the culture medium and washed the SH-SY5Y cells with PBS twice. Next, we added 200 μl of cold (4 °C) RIPA (P0013B; Beyotime Institute of Biotechnology) buffer for cell lysis. The lysate was centrifuged and the supernatant was harvested for protein concentration standardization. The subsequent experimental procedures were the same as the animal experiments.

Statistical analysis

All values are presented as means ± SEM. A two-tailed unpaired student t-test was used to compare the two groups; for multiple groups, we applied the one-way ANOVA followed by Bonferroni’s post-hoc tests. Probability values of p < 0.05 were regarded as statistically significant. All statistical analyses were performed using SPSS 24 software.

Results

Expression of HDAC6 and acetylated α-tubulin after ICH induction in vivo and in vitro

For the in vivo experiment, we detected the time course of the protein level of HDAC6 and acetylated α-tubulin from post-ICH perihematomal tissue via Western blot assay. As shown in Figs. 1A and 1B, HDAC6 dramatically increased starting 1 d post-ICH and reached peak at 3 d compared with the sham-treated group. In line with the in vivo result, the expression of HDAC6 in cultured SH-SY5Y cells was significantly up-regulated after hemin treatment for 24 h (Fig. 2B). As a main deacetylation substrate of HDAC6, acetylated α-tubulin decreased significantly in correlation with the in vivo and in vitro expression of HDAC6 (Figs. 1C and 2C). Meanwhile, we observed the localization of HDAC6 in the brain at 3 d after ICH. HDAC6 was primarily distributed in the cytoplasm of neurons. We barely found HDAC6 co-localization with astrocytes and microglia (Fig. 3).

Figure 1 Time-dependent trends of HDAC6 and acetylation α-tubulin protein levels in rats at acute intracerebral hemorrhage.

Western blot analysis (A), quantification analysis of HDAC6 (B), and acetylated α-tubulin (C) in an ICH animal model. Values are indicated by means ± SEM; vs. sham, *p < 0.05; **p < 0.01; ***p < 0.001.

Figure 2 HDAC6 and acetylation α-tubulin protein levels in SH-SY5Y cell models after ICH-induction at 24 and 48 h.

Western blot analysis (A), quantification analysis of HDAC6 (B), and acetylated α-tubulin (C) in an ICH cell model. Values are indicated by means ± SEM; vs. Cont, *p < 0.05; **p < 0.01; ***p < 0.001.

Figure 3 Double staining of HDAC6 with neuronal nuclei (NeuN), calcium-binding adaptor molecule 1 (IBA-1) and glial fibrillary acidic protein (GFAP) at 3 d after ICH.

n = 3. Scale bar = 20 μm.

Neuroprotection of HDAC6 inhibition in vivo and in vitro

After ICH induction, the rats displayed severe neurological impairments at 1 d and 3 d post-ICH, as measured by modified Garcia test (vs. sham: p < 0.001, Fig. 4A). Administration of a high dose of TubA (40 mg/kg) significantly improved behavior deficits between day 1 and day 3 post-ICH, while a medium dose of TubA (25 mg/kg) improved neurological deficits only on day 3. In line with the behavioral outcomes, H&E staining of perihematomal tissues exhibited that TubA treatment ameliorated histological damages at concentrations of 25 and 40 mg/kg (Fig. 4B). Compared with the TubA treatment groups, the vehicle group showed obvious tissue edema and more hyperchromatic cells with pyknotic nuclei. The influence of HDAC6 inhibition on SH-SY5Y cell viability was assessed using CCK-8. We found that the viability of hemin-induced cells was significantly inhibited compared with cells treated with TubA (3 μm), and specific HDAC6 knockdown by siRNA transfection significantly increased the viability of the cells (Fig. 4C). At the same time, the cell density increased under light microscope after HDAC6 inhibition.

Figure 4 Influences of HDAC6 inhibition treatment on ICH induction injuries.

(A) The neurological function assessed by a modified Garcia test in rats. Values are indicated by means ± SEM; vs. sham, *p < 0.05; **p < 0.01; ***p < 0.001. vs. ICH+Vehicle(DMSO), #p < 0.05; ##p < 0.01; ###p < 0.001. ICH+TubA (25 mg/kg) vs. ICH+TubA (40 mg/kg), ^p < 0.05; ^^p < 0.01; ^^^p < 0.001. (B) The histological changes evaluated via H&E staining at 3 d after ICH in rats. Scale bar = 50 and 10 μm. (C) Representative SH-SY5Y cells light photomicrographs and cellular viability assessed by the CCK8 assay in SH-SY5Y cells after 24 h treatment with hemin (10 μM). vs. Cont, *p < 0.05; **p < 0.01; ***p < 0.001. vs. Hemin+Vehicle, #p < 0.05; ##p < 0.01 ###p < 0.001. vs. Hemin+TubA, ^p < 0.05; ^^p < 0.01; ^^^p < 0.001. vs. Hemin+NC-siRNA, +p < 0.05; ++p < 0.01; +++p < 0.001.

HDAC6 inhibition attenuated ICH-induced brain edema

The BWC in the ipsilateral basal ganglia of ICH group notably increased when compared with the sham group at 1 d and 3 d after ICH induction (p < 0.001, Fig. 5). TubA, the HDAC6 inhibitor, did play a vital role in reducing cerebral edema at concentrations of 25 and 40 mg/kg (vehicle vs. TubA 25 or 40 mg/kg: p < 0.001). In addition, the effect of the high-dose group on brain edema was better compared to the low-dose group (TubA 40 mg/kg vs. TubA 25 mg/kg: p = 0.001, 1 d; p < 0.001, 3 d).

Figure 5 Brain water content assessed by the dry-wet weight way at 1 d and 3 d post-ICH.

Cont, contralateral; Ip, ipsilateral; BG, basal ganglia; CX, Cortex; Cerebel, cerebellum. vs. sham, *p < 0.05; **p < 0.01; ***p < 0.001. vs. ICH+Vehicle (DMSO), #p < 0.05; ##p < 0.01; ###p < 0.001. ICH+TubA (25 mg/kg) vs. ICH+TubA (40mg/kg), ^p < 0.05; ^^p < 0.01; ^^^p < 0.001.

HDAC6 inhibition attenuated ICH-induced neuronal apoptosis in vivo and in vitro

For the in vivo test, double immunofluorescent staining of TUNEL with NeuN was conducted in order to determine the number of apoptotic neurons after ICH (Fig. 6A). Apoptosis neurons were barely found in the sham-treated group. The percentage of apoptotic neurons in the total NeuN-positive cells was analyzed. As shown in Fig. 6B, a large quantity of apoptotic neurons in the ICH groups was observed, whereas TUNEL-positive neurons in the perihematomal tissue of the TubA-treated group (25 or 40 mg/kg) significantly decreased (p < 0.001). To further confirm whether HDAC6 inhibition also affected neuronal apoptosis in the in vitro models, we introduced flow cytometry analysis to count the number of apoptotic cells. The percentage of apoptotic SH-SY5Y cells increased after ICH induction. However, SH-SY5Y cells treated with TubA or transfected with HDAC6 siRNA both alleviated ICH-induced neuronal apoptosis. (p < 0.001, Figs. 7A and 7B).

Figure 6 Influences of HDAC6 inhibition treatment on neuronal apoptosis in ICH-induced rats.

(A) Representative microphotographs of TUNEL-positive neurons in perihematomal tissue at 3 d post-ICH. Scale bar = 20 μm. (B) Quantitative analysis (means ± SEM) of apoptotic neurons; vs. sham, *p < 0.05; **p < 0.01; ***p < 0.001. vs. ICH+Vehicle (DMSO), #p < 0.05; ##p < 0.01; ###p < 0.001. ICH+TubA (25 mg/kg) vs. ICH+TubA (40mg/kg), ^p < 0.05; ^^p < 0.01; ^^^p < 0.001.

Figure 7 Influences of HDAC6 inhibition treatment on neuronal apoptosis in Hemin-induced SH-SY5Y cells.

(A) Cell apoptosis assessed by cytometric analyses in SH-SY5Y cells after 24 h hemin (10 μM) induction. (B) Quantitative analysis (means ± SEM) of apoptotic cells by flow cytometric analyses. vs. Cont, *p < 0.05; **p < 0.01; ***p < 0.001. vs. Hemin+Vehicle, #p < 0.05; ##p < 0.01; ###p < 0.001. vs. Hemin+TubA, ^p < 0.05; ^^p < 0.01; ^^^p < 0.001. vs. Hemin+NC-siRNA, +p < 0.05; ++p < 0.01; +++p < 0.001.

HDAC6 inhibition suppressed the expression of apoptosis proteins and increased the expression of acetylated α-tubulin

To investigate whether the protein levels of HDAC6 were associated with neuronal apoptosis, immunoblotting was employed to test the protein levels of acetylated α-tubulin, Bax, cleaved caspase-3, and bcl-2 in each group. For the in vivo experiment, a medium dosage of TubA (25 mg/kg) and high dosage of TubA (40 mg/kg) significantly decreased HDAC6 and increased α-tubulin acetylation post-ICH induction (vehicle vs. TubA 25 mg/kg, p = 0.027; vehicle vs. TubA 40 mg/kg, p < 0.001, Figs. 8A and 8B). Moreover, the expression levels of Bax and cleaved caspase-3 significantly decreased, and Bcl-2 significantly increased in the TubA treatment groups (Figs. 8D–8G). For the in vitro experiment, the above results confirmed again that TubA treatment or HDAC6 knockdown attenuated neuronal death by decreasing Bax and cleaved caspase-3 in hemin-induced SH-SY5Y cells (Fig. 9). In general, these results strongly indicate that decreased HDAC6 expression ameliorated ICH-induced neuronal apoptosis and that acetylated α-tubulin may play a potential role in this pathophysiological process.

Figure 8 Inhibition of HDAC6 confers neuronal protection against ICH-induced α-tubulin deacetylation and apoptosis protein generation in rats.

Western blot analysis (A), quantification analysis of HDAC6 (B), and acetylated α-tubulin (C). Western blot analysis (D) and quantification analysis (means ± SEM) of Bax (E), bcl-2 (F), and cleaved caspase-3 (G). vs. sham, *p < 0.05; **p < 0.01; ***p < 0.001. vs. ICH+Vehicle(DMSO), #p < 0.05; ##p < 0.01; ###p < 0.001. ICH+TubA (25 mg/kg) vs. ICH+TubA (40 mg/kg), ^p < 0.05; ^^p < 0.01; ^^^p < 0.001.

Figure 9 Inhibition of HDAC6 confers neuronal protection against ICH-induced a-tubulin deacetylation and apoptosis proteins generation in SH-SY5Y cells.

Western blot analysis (A), quantification analysis of HDAC6 (B), and acetylated α-tubulin (C). Western blot analysis (D) and quantification analysis (means ± SEM) of Bax (E), bcl-2 (F) and cleaved caspase-3 (G). vs. Cont, *p < 0.05; **p < 0.01; ***p < 0.001. vs. Hemin+Vehicle, #p < 0.05; ##p < 0.01; ###p < 0.001. vs. Hemin+TubA, ^p < 0.05; ^^p < 0.01; ^^^p < 0.001. vs. Hemin+NC-siRNA, +p < 0.05; ++p < 0.01; +++p < 0.001.

Discussion

In this study, we investigated the role of HDAC6 in early brain injury after ICH and explored its underlying molecular mechanisms. The main observations are as follows: (1) the collagenase-induced ICH model enhanced the expression of HDAC6, which dramatically increased 1 d after ICH, and peaked at 3 d. In the cellular model of ICH, the levels of HDAC6 in SH-SY5Y cells also increased significantly after hemin treatment for 24 h. On the other hand, the acetylated α-tubulin significantly decreased in parallel with HDAC6 expression in the cellular and animal models of ICH; (2) HDAC6 inhibition by TubA at both medium and high doses (25, 40 mg/kg) improved neurological deficits, histological impairments, cerebral edema, and perihematomal neuronal apoptosis in the ICH animal model; (3) for the in vitro experiment, HDAC6 inhibition by TubA or siRNA inhibited apoptosis of SH-SY5Y cells after hemin treatment for 24 h; and (4) both animal and cellular experiments confirmed that HDAC6 inhibition increased the acetylated α-tubulin and Bcl-2 while reducing the expression of Bax and cleaved caspase-3 after ICH. In general, these results hint that HDAC6 inhibition might be a promising therapeutic strategy for ICH treatment.

HDAC6 is a unique member of the class IIb HDACs with an ubiquitin-interacting domain and two catalytical domains (Hubbert et al., 2002). This HDAC family regulates many critical cellular biological courses, involving cell migration, immune synapse formation, microtubule-based transport, aggresome generation, and autophagy via deacetylating non-histone proteins like α-tubulin, β-catenin, and HSP 90 (Valenzuela-Fernández et al., 2008). It is a prospective therapy target used against the neuroprotection and regeneration of CNS diseases (Rivieccio et al., 2009). For decades, high selective inhibitors or small interfering RNAs targeting HDAC6 exhibited neuroprotection in several experimental models of neurological disorders. Previous studies showed it can rescue cortical neuron necroptosis in an oxygen-glucose deprivation model (Yuan et al., 2015), attenuate cerebral infarction size and function deficit in an ischemia stroke model (Wang et al., 2016), improve cognitive impairments in an animal model of Alzheimer’s disease (Zhang et al., 2014), and promote functional recovery in a cauda equina injury model (Fu et al., 2018). Although the neuroprotective capacity of HDAC6 inhibition has been studied for years, whether HDAC6 inhibition had an analogous neuroprotective effect on ICH remained unclear.

Hence, in the first part of this study, we explored the expression and location of HDAC6 in cellular and animal models of ICH via Western blot and immunofluorescence staining. Chen et al. (2012) demonstrated that HDAC6 expression significantly increased during the early phases of ischemia stroke, indicating its contribution to stroke pathogenesis. Zhang et al. (2014) noticed a dramatic increase in the level of HDAC6 in a transgenic mice model of Alzheimer’s disease. In line with these results, our findings demonstrated that the level of HDAC6 increased significantly at 1 day and peaked at 3 days after ICH induction. For the in vitro experiment, the level of HDAC6 in SH-SY5Y cells significantly increased after hemin treatment for 24 h. In addition, the morphological study confirmed that HDAC6 was predominantly distributed in neurons, not astrocytes or microglia, which implied that it might play a significant role in determining the survival of neurons.

In view of the observations mentioned above, we then studied the neuroprotection of HDAC6 inhibition on collagenase-induced rats and hemin-induced SH-SY5Y cells. Our results demonstrated that both medium and high doses of TubA (25, 40 mg/kg) improved neurological deficits after ICH induction and relieved ipsilateral cerebral edema during the acute stages of ICH. Additionally, HE staining confirmed that HDAC6 inhibition can improve histological impairments after ICH induction. On the other hand, we demonstrated that HDAC6 inhibition by TubA or siRNA can improve the viability rate of SH-SY5Y cells after hemin treatment. Ultimately, this is the first article to confirm the neuroprotective effects of HDAC6 inhibition in an ICH cellular and animal model.

The underlying neuroprotective mechanisms of HDAC6 inhibition have yet to be clarified. As a classic histone deacetylase (HDAC) inhibitor, TubA provides a variety of functions including preventing mitochondrial dysfunctions (Dompierre et al., 2007) and restoring endothelial barrier dysfunction (Yu et al., 2016) and anti-necroptosis (Yuan et al., 2015) by increasing acetylated α-tubulin. Our data showed that the acetylation levels of α-tubulin significantly decreased along with the increase of HDAC6 in ICH models, and the down-regulation of acetylated α-tubulin was ameliorated by HDAC6 inhibitor TubA. In addition, our previous research demonstrated that inhibiting HDAC6 using tubacin or specific shRNA upregulated acetylated α-tubulin and inhibited subsequent apoptosis by preventing OGDR-induced Golgi fragmentation (Zhang et al., 2019). Hence, we also investigated the influence of HDAC6 inhibition on neuronal apoptosis in the ICH model. The results showed that TubA or HDAC6 knockdown significantly reduced the neuronal apoptosis rate, levels of cleaved caspase-3 and Bax, and increased the expression of Bcl-2. We also confirmed that HDAC6 inhibition can improve neuronal apoptosis in an ICH model. A recent study by Wang et al. (2022) found that HDAC6 knockout can up-regulate the acetylation of MDH1, which may play an anti-apoptotic role by alleviating oxidative stress in ICH. Our research further confirms the ability of pharmacological inhibitor TubA to up-regulate α-tubulin acetylation and its anti-apoptotic effect in neurons. We also compared the effects between different doses of TubA in vivo. Apoptosis manifested as obvious morphologic changes, shrinkage of the neuronal cell body, condensation of chromatin, fragmentation of neuronal nucleus, and a critical role in the pathological course of secondary brain injury after ICH (Bobinger et al., 2018). As a primary substrate of HDAC6, the expression level of acetylated α-tubulin can maintain microtubule-based transport between the dendrites/axon and neuronal cell body, as well as improve outcomes in CNS injury (Wang et al., 2016). Taking all into consideration, the up-regulation of α-tubulin acetylation and inhibition of neuronal apoptosis made an important contribution for selective HDAC6 inhibition-mediated protective effects following ICH. Finally, the relationship between the upregulation of acetylated α-tubulin and anti-apoptosis was not studied, but will be explored in in-depth follow-up experiments.

Conclusion

The results of our research suggested that HDAC6 played a vital role in the pathological course of early brain injury after ICH. We further demonstrated that inhibiting the expression of HDAC6 by TubA or specific siRNA had beneficial effects after experimental ICH. Both in vivo and in vitro experiments verified that the protective effects may involve up-regulating the acetylation of α-tubulin and reducing neuronal apoptosis. These results suggested that the inhibition of HDAC6 could be a novel and promising therapeutic target for ICH therapy.

Supplemental Information

Supplemental Information 1 Rats Assignment and Use.

Garcia, Garcia test; WB, western blot; IF, immunofluorescence staining; BWC, brain water content; TUNEL, TdT-mediated dUTP-biotin nick end labeling staining; HE, Hematoxylin and eosin staining.

Click here for additional data file.

Supplemental Information 2 HDAC6 siRNA sequences.

Click here for additional data file.

Supplemental Information 3 Raw WB gel pictures.

Click here for additional data file.

Supplemental Information 4 Raw statistical data.

Click here for additional data file.

Supplemental Information 5 ARRIVE 2.0 Checklist.

Click here for additional data file.

Additional Information and Declarations

Competing Interests

Author Contributions

Animal Ethics

Data Availability

The authors declare that they have no competing interests.

Cuiying Peng conceived and designed the experiments, performed the experiments, analyzed the data, prepared figures and/or tables, authored or reviewed drafts of the article, and approved the final draft.

Xiyu Gong analyzed the data, prepared figures and/or tables, and approved the final draft.

Zhiping Hu analyzed the data, authored or reviewed drafts of the article, and approved the final draft.

Chunli Chen conceived and designed the experiments, performed the experiments, analyzed the data, authored or reviewed drafts of the article, and approved the final draft.

Zheng Jiang conceived and designed the experiments, performed the experiments, analyzed the data, authored or reviewed drafts of the article, and approved the final draft.

The following information was supplied relating to ethical approvals (i.e., approving body and any reference numbers):

All experimental programs were authorized by the Institutional Animal Care and Use Committee of Central South University (IACUC approval No: 2020079)

The following information was supplied regarding data availability:

The raw measurements are available in the Supplemental Files.

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
