# Peer review of "Selective HDAC6 inhibitor TubA offers neuroprotection after intracerebral hemorrhage via inhibiting neuronal apoptosis"

_PeerJ, doi:10.7717/peerj.15293_

## Round 0.1 · original submission · Major Revisions

This manuscript is similar to previously published studies. In addition, the manuscript contains some grammatical errors.

Reviewer 1 ·

Basic reporting

1) Some typo errors need to be checked, such as phamacological→pharmacological (line 42); favourable or favorable (line 83, I think American English writing was used by the authors in this article.); a-tubulin→alpha-tubulin (lines 144, 252, 254, ……), etc.

Experimental design

1) According to the Rats-used group and Rats-died group mentioned in the supplementary material 1 file, the total number of Rats used in this experiment is over one hundred and seventy (lines 102-103).

2) In the time-dependent experiment, fifteen sham-operated rats were used but three sham-operated rats were involved in the results. The authors need to have a reason for the other twelve sham-operated rats (Figure 1).

Validity of the findings

1) An article published in 2022 found that inhibition of HDAC6 by siRNA or Tubastatin A treatment was neuroprotective after ICH (Upregulation of MDH1 acetylation by HDAC6 inhibition protects against oxidative stress‑derived neuronal apoptosis following intracerebral hemorrhage. Miao Wang et al., Cell Mol Life Sci. 2022 Jun 9;79(7):356. doi: 10.1007/s00018-022-04341-y.). This reference did not be mentioned in this submitted article. Moreover, this submitted article has similar results compared to the previous study (Miao Wang et al.). The authors should indicate the difference or new findings of this submitted article.

2) Does acetylation of alpha-tubulin directly participate in mediating apoptotic mechanism or just an outcome after HDAC6 inhibition? The result is still not clear after the description by the authors in the introduction part (lines 90-93) and discussion part (line 379).

Additional comments

No additional comments.

Reviewer 2 ·

Basic reporting

This study has a clear focus on exploring the role of HDAC6 in the context of intracerebral hemorrhage (ICH) and the potential protective effects of TubA on apoptosis in an ICH model. By examining these specific aspects, the study aims to contribute to the current understanding of the mechanisms underlying ICH and potentially identify new therapeutic strategies for this condition.

Experimental design

no comment.

Validity of the findings

There have been numerous studies on the relationship between HDAC6 and apoptosis, making it difficult for a new study on the same topic to be considered highly innovative. Additionally, the study need bring together and synthesize existing knowledge in a unique way, or present findings from a different perspective, making it valuable and relevant to the scientific community.

Additional comments

The author should shorten the abstract. A well-written abstract should succinctly provide an overview of the study, including its purpose, methodology, results, and conclusions. The aim is to provide readers with a clear understanding of the research without going into excessive detail.
It's possible that the study may not be considered highly innovative, and in such cases, further research or refinement may be needed before it can be considered for publication.

Reviewer 3 ·

Basic reporting

The article is fundamentally flawed and should be rejected without possibility of revision.

Experimental design

The article is fundamentally flawed and should be rejected without possibility of revision.

Validity of the findings

The article is fundamentally flawed and should be rejected without possibility of revision.

Additional comments

The article is fundamentally flawed and should be rejected without possibility of revision.

---

## Round 0.2 · accepted · Accept

The authors have well addressed all of the reviewers' comments.

Reviewer 1 ·

Basic reporting

No further questions

Experimental design

No further questions

Validity of the findings

No further questions

Additional comments

No additional comments

Reviewer 2 ·

Basic reporting

no comment

Experimental design

no comment

Validity of the findings

no comment

Additional comments

no comment

Reviewer 3 ·

Basic reporting

This study well elucidates the role of selective HDAC6 inhibitor TubA in neuroprotection after intracerebral hemorrhage and provides new experimental evidence to support this effect. The results have clinical therapeutic potential and provide a theoretical basis for further development of novel drugs for cerebral hemorrhage treatment.

Experimental design

The structure of the paper is clear, the experimental design is reasonable, the experimental data are sufficient, and the figures and tables are clear and readable.

Validity of the findings

The Results section of this study provides a clear description of the experimental data and provides sufficient evidence to support the role of selective HDAC6 inhibitor TubA in neuroprotection.

Additional comments

In summary, this paper demonstrates excellent performance in experimental design, data analysis, and conclusion elaboration